# Nickel-catalyzed allylic carbonylative coupling of alkyl zinc reagents with *tert*-butyl isocyanide

Yangyang Weng[1], Chenhuan Zhang[1], Zaiquan Tang[1], Mohini Shrestha[1], Wenyi Huang[1], Jingping Qu[1]* & Yifeng Chen [1]*

Transition metal-catalyzed carbonylation with carbon nucleophiles is one of the most prominent methods to construct ketones, which are highly versatile motifs prevalent in a variety of organic compounds. In comparison to the well-established palladium catalytic system, the nickel-catalyzed carbonylative coupling is much underdeveloped due to the strong binding affinity of CO to nickel. By leveraging easily accessible *tert*-butyl isocyanide as the CO surrogate, we present a nickel-catalyzed allylic carbonylative coupling with alkyl zinc reagent, allowing for the practical and straightforward preparation of synthetically important β,γ-unsaturated ketones in a linear-selective fashion with excellent *trans*-selectivity under mild conditions. Moreover, the undesired polycarbonylation process which is often encountered in palladium chemistry could be completely suppressed. This nickel-based method features excellent functional group tolerance, even including the active aryl iodide functionality to allow the orthogonal derivatization of β,γ-unsaturated ketones. Preliminary mechanistic studies suggest that the reaction proceeds via a π-allylnickel intermediate.

---

[1] Key Laboratory for Advanced Materials and Joint International Research Laboratory of Precision Chemistry and Molecular Engineering, Feringa Nobel Prize Scientist Joint Research Center, School of Chemistry and Molecular Engineering, East China University of Science and Technology, 130 Meilong Road, Shanghai 200237, China. *email: qujp@dlut.edu.cn; yifengchen@ecust.edu.cn

Pioneering work developed by Heck in the 1970s catapulted palladium (Pd)-catalyzed three component reactions with carbon monoxide (CO) as a powerful strategy for introduction of carbonyl group[1–8]. Among these processes, the use of carbon-based nucleophiles allows for the convenient synthesis of ketones with broad applications[9]. Pd-catalyzed allylic reaction represents one of the most prominent carbon–carbon bond-forming reactions with wide synthetic applications in organic chemistry, including synthesis of various biologically active natural products, pharmaceuticals, and agrochemicals[10–13]. Thus, the allylic carbonylation would be an important strategy for incorporation of both alkene and carbonyl functionality in one synthetic protocol, enabling the expedient synthesis of the versatile β,γ-unsaturated ketones, which are ubiquitous motifs in bioactive compounds and utilized as valuable synthetic building blocks[14–20]. The Stille group has realized the Pd-catalyzed allylic carbonylative Stille coupling, while the organotin reagents largely limited on aryl, vinyl, and allyl stannanes[21,22]. The Tamaru group has previously developed a Pd-catalyzed allylic carbonylative Negishi coupling with CO to access β,γ-unsaturated ketones[23,24]. However, the site selectivity of this process is highly dependent on the electronics of the organozinc coupling partner, and a mixture of the linear and branched coupling products were usually obtained when simple alkyl Negishi reagents were used (Fig. 1a). Despite the progress in Pd-catalyzed three component reactions with CO gas, it is still highly imperative to develop practical carbonylation utilizing earth abundant transition metal[25,26] for functionalized ketone synthesis, especially to circumvent the long-term limitations.

Isocyanide is an important array of organic reagent widely used in transition metal catalyzed carbonylations as C-1 source and in heterocycle synthesis[27–38]. Despite the progress, the leverage of functionalized alkyl Negishi reagents for transition metal catalyzed carbonylative coupling with isocyanide still remains extremely limited[33]. Recently, Dechert-Schmitt and Blackmond reported a modular unsymmetrical 1,2-diketone synthesis via Pd-catalyzed four-component coupling between an aryl halide, an alkyl zinc reagent and two molecules of $^t$BuNC. In this process,

resting bisiminoyl Pd intermediate was formed between the Pd-iminoacyl species and 1-iminoalkyl zinc reagents (Fig. 1b)[33].

At the outset of our investigation, we recognized several issues that needed to be addressed in order to develop an effective nickel-catalyzed allylic carbonylation. First, the use of nickel catalysts in carbonylation reactions has been less explored likely due to the strong binding affinity of CO towards nickel[39–53]. As a rare example, the Skrydstrup group elegantly succeeded the catalytic carbonylation of primary benzylic electrophile using a nickel pincer complex with a CO-gen precursor via the slow addition of Negishi reagent to circumvent the direct Negishi coupling, although the use of allyl electrophiles received limited success with 8% ketone formation[51]. Additionally, the preference of imidoylnickel intermediates to undergo further migratory insertion with isocyanides to furnish poly(iminomethylene)s is a well-known process in polymer chemistry[54–57], presenting a major hurdle to access monocarbonylated products. Furthermore, Zhu and co-workers[58] illustrate that the allyl imidoylpalladium intermediate is extremely prone to undergo β-H elimination to provide the ketenimine intermediate, which could further hydrolyze to β,γ-unsaturated amide[59]. To overcome these challenges, we envisioned that the use of functional group compatible organozinc reagents that are highly effective for transmetallation might disfavor the overcarbonylation process. Rapid C–C bond formation via reductive elimination would provide the mono-carbonylated coupling product[60,61]. Moreover, the β,γ-unsaturated ketone easily undergoes the undesired isomerization step to afford the thermodynamically stable α,β-unsaturated ketone in the presence of transition metal catalyst[62]. Herein, we report the highly regio- and chemoselective nickel-catalyzed allylic carbonylative Negishi reaction with tert-butyl isocyanide[63], which allows the expedient synthesis of β,γ-unsaturated ketones with broad substrate scope under mild conditions (Fig. 1c).

## Results

**Optimization of reaction conditions.** We started our investigation by studying the nickel-catalyzed reaction of phenyl allyl

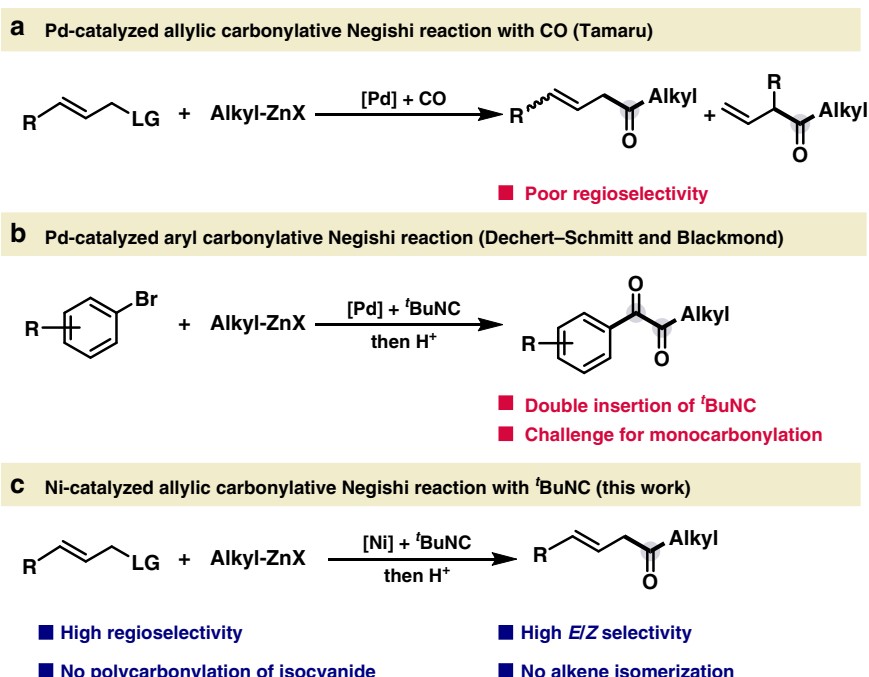

**Fig. 1 Development of transition metal-catalyzed carbonylative Negishi reaction. a** Pd-catalyzed allylic carbonylative Negishi reaction. **b** Pd-catalyzed aryl carbonylative Negishi reaction. **c** Ni-catalyzed allylic carbonylative Negishi reaction.

### Table 1 Optimization of the reaction conditions.

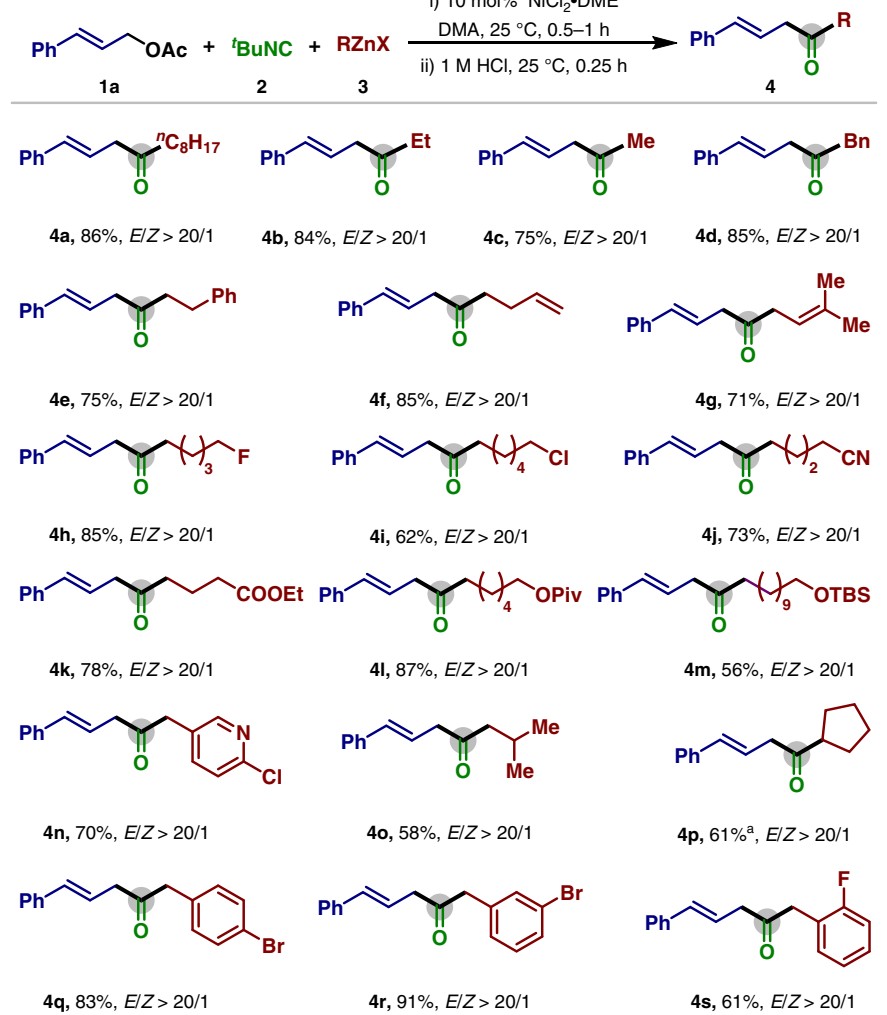

| Entry[a] | Catalyst | Ligand | Yield (%)[b] | | |
|---|---|---|---|---|---|
| | | | 4a | 5 | 6 |
| 1 | NiCl₂·DME | — | 90 (86)[c] | ND | 1 |
| 2 | NiCl₂·DME | PPh₃ | 27 | 10 | ND |
| 3 | NiCl₂·DME | dppf | 5 | ND | ND |
| 4 | NiCl₂·DME | Phen | 82 | 3 | 3 |
| 5 | NiCl₂·DME | SIPr·HCl | 87 | 2 | 3 |
| 6 | Ni(cod)₂ | — | 86 | 6 | ND |
| 7 | Pd(dba)₂ | — | 4 | ND | ND |
| 8 | — | — | ND | ND | ND |
| 9[d] | NiCl₂·DME | — | ND | ND | ND |

*dppf* 1,1′-Ferrocenediyl-bis(diphenylphosphine), *SIPr·HCl* 1,3-Bis-(2,6-diisopropylphenyl)imidazolium chloride, *ND* not determined
[a]Reaction conditions: **1a** (0.2 mmol), **2** (0.3 mmol), **3a** (0.3 mmol), catalyst (0.02 mmol), ligand (0.04 mmol), DMA (2 mL), 25 °C, 0.5 h, and then 1 M HCl (2 mL) was added, 25 °C, 0.25 h
[b]Corrected GC yield
[c]Isolated yield within parentheses in 0.2 mmol scale
[d]The reaction in CO (1 atm) instead of ᵗBuNC

**Fig. 2 Scope of alkyl zinc reagent for nickel-catalyzed allylic carbonylative Negishi reaction.** Reaction conditions: **1a** (0.2 mmol), **2** (0.3 mmol), **3** (0.3 mmol), NiCl₂·DME (0.02 mmol), DMA (2 mL), 25 °C, 0.5 h, and then 1 M HCl. [a]Ni(cod)₂ (0.02 mmol), 50 °C.

**Fig. 3 Scope of allyl acetate for nickel-catalyzed allylic carbonylative Negishi reaction.** Reaction conditions: **1** (0.2 mmol), **2** (0.3 mmol), **3** (0.3 mmol), NiCl$_2$·DME (0.02 mmol), DMA (2 mL), 25 °C, 0.5 h, and then 1 M HCl. [a]50 °C; [b]Ni(cod)$_2$ (0.02 mmol); [c]the ratio of regioisomer is 6:1; [d]the ratio of regioisomer is 4:1.

acetate **1a** and $^{n}$C$_8$H$_{17}$ZnBr (1.5 equiv). To our delight, when the commercially available *tert*-butyl isocyanide (1.5 equiv) was used as the CO equivalent, the reaction proceeded smoothly with 10 mol% bench-stable and easily accessible NiCl$_2$·DME in dimethylacetamide (DMA) at 25 °C. The starting material was consumed in 30 min; affording the desired *trans* β,γ-unsaturated ketone **4a** in 86% isolated yield after the simple acidic work up procedure, only small amount of direct coupling byproduct **6** (1%) could be observed at gas chromatography (GC) (Table 1, entry 1). This nickel-isocyanide system features excellent linear selectivity. Neither the branched nor the secondary allylic products as observed in previous work[23] were detected. Ligand-free nickel species was found to be the most effective for this coupling process. The use of phosphine ligands including PPh$_3$ and other bidentate phosphines such as dppf (1,1′-ferrocenediyl-bis(diphenylphosphine)) was detrimental to this reaction, β,γ-unsaturated amide byproduct **5** was detected by GC (Table 1, entries 2 and 3). The use of 1,10-phenathorine also lowered the yield (Table 1, entry 4), and the inclusion of N-heterocyclic carbene ligand was also not beneficial (Table 1, entry 5). Additionally, Ni(0) precatalysts such as Ni(cod)$_2$ provided similar results (Table 1, entry 6). Interestingly, Pd(dba)$_2$ only afforded 4% ketone product (**4a**), demonstrating the difference in reactivity between the two transition metals (Table 1, entry 7). Finally, control experiments showed that the nickel catalyst was essential for this three-component reaction (Table 1, entry

8). The use of CO gas was also ineffective for this coupling (Table 1, entry 9).

**Substrate scope of alkyl zinc reagents.** With the optimized conditions in hand, we next explored the substrate scope of alkylated zinc nucleophiles (Fig. 2). The reaction tolerates a wide range of organozinc reagents, affording β,γ-unsaturated ketones with complete *trans*-selectivity. Furthermore, the undesired isomerization of alkene moiety was not detected. Diethyl zinc could be successfully applied in this carbonylation process with 84% isolated yield (**4b**); however, requiring sacrifice one equivalent alkyl source. Simple alkyl groups including methyl (**4c**), benzyl (**4d**), and 2-phenylethyl (**4e**) could be incorporated into the products in excellent yield. The alkene moiety was compatible in this carbonylative process, and homoallyl (**4f**) and prenyl (**4g**) substituted β,γ-unsaturated ketones could be formed in good yield with excellent regio- and stereoselectivity. Additionally, various functionalized alkylzinc reagents, including those possessing a fluoride (**4h**), a chloride (**4i**), a nitrile (**4j**), an ester (**4k**), and an ether (**4l**, **4m**), were successfully converted to the corresponding ketone product. Moreover, secondary cyclopentyl zinc bromide could also serve as the nucleophile to afford **4p** in moderate yield. The utilization of nickel catalyst also tolerated the halogen-containing (hetero)benzyl nucleophiles (**4n** and **4q**–**4s**), allowing further functionalization to be carried out. Unfortunately, the use of the tertiary alkylzinc and arylzinc reagent did

**Fig. 4 Scope of biologically active allyl acetate for nickel-catalyzed allylic carbonylative Negishi reaction.** Reaction conditions: **1** (0.2 mmol), **2** (0.3 mmol), **3** (0.3 mmol), NiCl$_2$·DME (0.02 mmol), DMA (2 mL), 25 °C, 0.5 h, and then 1 M HCl. [a]Ni(cod)$_2$ (0.02 mmol), 50 °C.

not afford the desired carbonylative cross-coupling product under these conditions.

**Substrate scope of allylic electrophiles**. The scope of allylic electrophiles was also investigated, the reaction proceeded well with both aryl and alkyl substituted alkene, affording the carbonylation product in 43–96% yield with excellent regio- and stereoselectivities (Fig. 3). Aryl substituents including 2-Br (**4v**), 4-I (**4w**), and 4-Bpin (**4x**) were tolerated, providing the feasibility for subsequent derivatization. The employment of heteroaromatic ring including thiophene (**4y**), furan (**4z**), and indole (**4aa**) also provided the product in high isolated yield. When a methyl group was introduced at the C-2 position of the allyl electrophile (**4ae**), the reaction also proceeded with excellent *E/Z* selectivity. The migratory insertion of allylic nickel intermediate with isocyanide mainly proceeded in the least sterically hindered position, furnishing the *trans* ketone **4af–4ah** in good yield.

To illustrate potential utility of this nickel-catalyzed three-component carbonylative coupling, we performed late-stage modifications on complex and/or biologically active compounds (Fig. 4). For instance, several pharmaceutical derivatives, naproxen (**4aj**), ibuprofen (**4ak**), hyodeoxycholic acid (**4am**), and indometacin (**4an**) could be readily applied in this reaction,

affording the desired product in good yield. To our delight, the olefin containing citronellal (**4ai**) and oleic acid (**4al**) both could be effectively transformed to relatively β,γ-unsaturated ketones in regioselective way, while no alkene isomerization could be observed in the standard condition. Notably, vitamin-E (**4ao**) derivative is also amenable and the desired product could be obtained in moderate yield. These results clearly demonstrate that this carbonylation has promising applications in late-stage modification of complex molecules.

**Synthetic applications**. To further showcase the synthetic potential for this nickel catalyzed carbonylative Negishi reaction using *tert*-butyl isocyanide as carbonyl source, the gram-scale synthesis of **4t** was carried out with 90% isolated yield (Fig. 5a). The imine intermediate could be reduced using NaBH$_4$ to provide the useful building block homoallylic amine **7** (83% yield), further demonstrating the advantage of this isocyanide coupling technology (Fig. 5b). The tolerance of active aryl halide under current conditions allows the orthogonal cross-coupling strategy (Fig. 5c). Pd-catalyzed Suzuki coupling of (*E*)-3-(4-iodophenyl)allyl acetate **1w** with phenylboronic acid afforded intermediate **1ap**, which could further undergo the allylic carbonylation with benzyl zinc reagent to provide β,γ-unsaturated ketones **4ap** in 58% overall yield. Meanwhile, β,γ-unsaturated ketones **4w** could be obtained

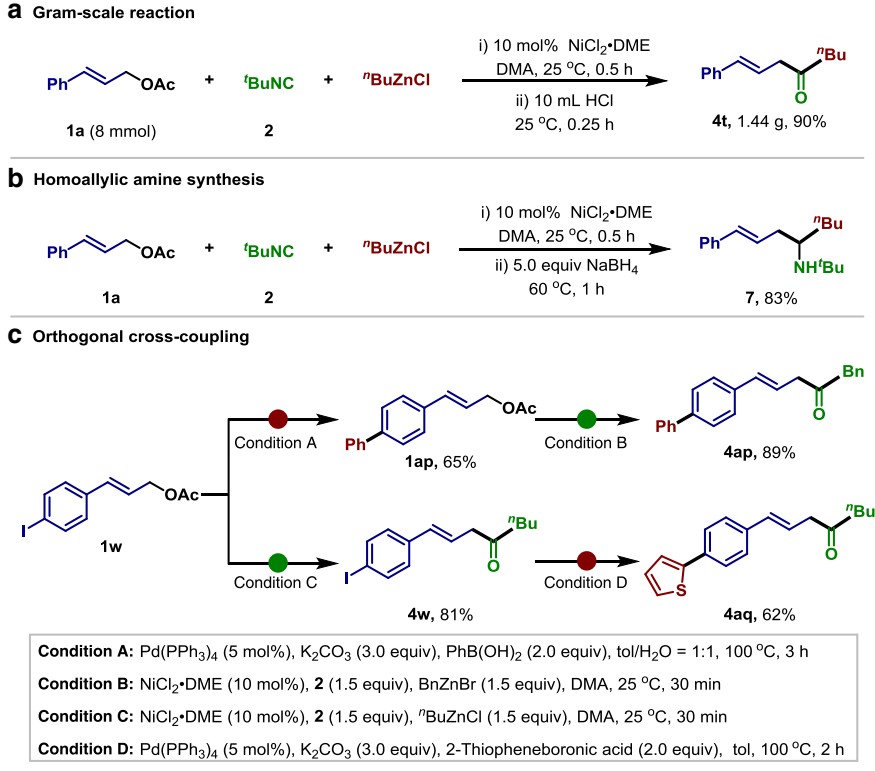

**Fig. 5 Synthetic application. a** Gram-scale experiment. **b** Synthesis of homoallylic amine. **c** Orthogonal cross-coupling.

via the selective Ni-catalyzed carbonylative Negishi reaction, and then thiophene functionality could be incorporated through the Suzuki coupling to deliver the desired product **4aq** in 51% yield.

**Mechanistic studies and proposed mechanism.** To shed light on the reaction mechanism, several experiments were performed (Fig. 6). When 1-phenylallyl acetate **1a′** was used as the allylic electrophile, the linear β,γ-unsaturated ketone **4t** was obtained as the sole product in 88% yield. Additionally, when allyl acetate containing a cyclopropyl group (**1ar**) was subjected to the standard conditions, ring-opening product of the cyclopropyl group could not be detected (Fig. 6a). Together, these results indicate that the reaction might proceed via the π-allylnickel intermediate, and the radical intermediates is unlikely involved. The stereochemistry of this carbonylative coupling reaction was also examined by using (R)-**1af** as the starting material (Fig. 6b): the desired product **4af** was obtained with 75% ee (enantiomeric excess) with the inversed configuration, which indicated that the initiate step most likely undergo SN2 type in the oxidative addition process similar to the Tsuji–Trost-type oxidation[64–66]. The erosion of ee may arise from the tautomerization between imine and enamine intermediate in the acidic work up procedure. The *tert*-butyl isocyanide possesses strong binding affinity to the nickel center, we prepared the tetrakis(*tert*-butyl isocyanide)nickel (II) perchlorate complex **8**. Under standard condition, using 0.38 equiv **8** in the absence of additional *t*BuNC delivered the desired product in 57% yield. In contrast, using 10 mol% **8** with the addition of 1.5 equiv *t*BuNC increased the yield to 82%. These results suggest that the binding of multiple *t*BuNC as ligands to the nickel catalyst may proceed with ligand dissociation process in the catalytic cycles and slow down the coupling process (Fig. 6c). When we preform the [1-(*tert*-butylimino)-butyl]zinc chloride, which is prepared in situ via the reaction of *t*BuNC

and *n*BuLi followed by transmetallation with ZnCl₂ at room temperature, the desired product **4t** was not observed under our conditions, in contrast to the previous findings of Ito and co-workers[67] and Dechert-Schmitt et al.[33], where high temperature is essential for carbonylative coupling (Fig. 6d). This result highlights the distinctive mechanism in the Ni-catalyzed carbonylative Negishi coupling reaction.

Based on the preliminary mechanistic studies, a plausible reaction pathway is proposed in Fig. 7. Oxidative addition of allyl acetate **1** with the nickel catalyst affords the π-allylnickel(II) intermediate **A**. Migratory insertion of *tert*-butyl isocyanide then provides allyl imidoylnickel intermediate **B**, from which transmetallation and reductive elimination lead to the β,γ-unsaturated imine **D**. The desired β,γ-unsaturated ketone **4** could be obtained via acidic hydrolysis of **D**.

## Discussion

In summary, nickel-catalyzed allylic carbonylative coupling with alkyl zinc reagent has been developed for the synthesis of β,γ-unsaturated ketones from allylic acetate and alkyl zinc reagent using commercially available *tert*-butyl isocyanide as a CO source. In this coupling process, the allyl imidoylnickel intermediate undergoes rapid transmetallation with the zinc nucleophile, thus avoiding the undesired polycarbonylation. This reaction features broad substrate scope with excellent regio- and chemoselectivity. Preliminary mechanistic studies reveal the reaction proceeds with the π-allyl nickel intermediate. Further effort on detailed mechanism and exploration of other electrophiles is currently underway in our laboratory will be reported in future.

## Methods

**General procedure A for the allylic carbonylative reaction.** An oven-dried Schlenk tube charged with NiCl₂·DME (10 mol%) was evacuated and backfilled with N₂. (This process was repeated for three times.) DMA (0.1 M) was added into

**a** The reaction proceeds with π-allyl nickel intermediate

[Reaction scheme: 1a′ + $^t$BuNC (2) + $^n$BuZnCl → i) 10 mol% NiCl$_2$•DME, DMA, 25 °C, 0.5 h; ii) 1 M HCl, 25 °C, 0.25 h → 4t, 88%]

[Reaction scheme: 1ar + $^t$BuNC (2) + $^n$BuZnCl → i) 10 mol% Ni(cod)$_2$, DMA, 50 °C, 1 h; ii) 1 M HCl, 25 °C, 0.25 h → 4ar, 61%]

**b** The carbonylative reaction used (R,E)-4-phenylbut-3-en-2-yl acetate

[Reaction scheme: (R)-1af, 99% ee + $^t$BuNC (2) + MeZnBr → i) 10 mol% Ni(cod)$_2$, DMA, 50 °C, 1 h; ii) 1 M HCl, 25 °C, 0.25 h → (S)-4af, 79%, 75% ee]

**c** The use of Ni ($^t$BuNC)$_4$(ClO$_4$)$_2$ as carbonyl source

[Reaction scheme: 1a + $^n$BuZnCl → i) 0.38 equiv Ni($^t$BuNC)$_4$(ClO$_4$)$_2$ (8), DMA, 25 °C, 0.5 h; ii) 1 M HCl, 25 °C, 0.25 h → 4t, 57%]

[Reaction scheme: 1a + $^n$BuZnCl → i) 10 mol % Ni($^t$BuNC)$_4$(ClO$_4$)$_2$ (8), 1.5 equiv $^t$BuNC (2), DMA, 25 °C, 0.5 h; ii) 1 M HCl, 25 °C, 0.25 h → 4t, 82%]

**d** The use of [1-(tert-butylimino) butyl] zinc reagent in the reaction

[Reaction scheme: $^t$BuNC + $^n$BuLi (2) → i) THF, −5 °C, 1 h; ii) 1.2 equiv ZnCl$_2$, −5 °C to 25 °C, 0.5 h; iii) 10 mol % NiCl$_2$•DME, 1.0 equiv cinnamyl acetate, DMA, 25 °C, 0.5 h → 4t, 0%]

**Fig. 6 Preliminary mechanistic studies. a** The reaction proceeds with π-allyl nickel intermediate. **b** Stereochemistry of this carbonylative coupling reaction. **c** The use of Ni($^t$BuNC)$_4$(ClO$_4$)$_2$ as carbonyl source. **d** The use of [1-(tert-butylimino) butyl] zinc reagent in the reaction.

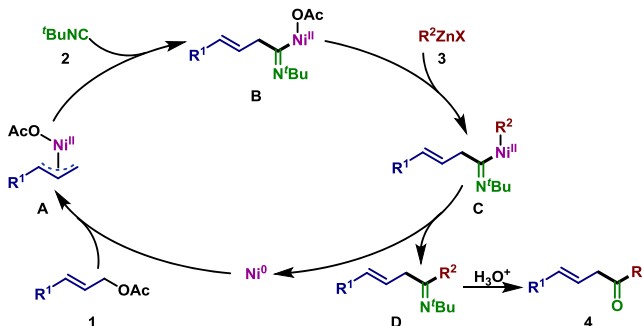

**Fig. 7 Tentative mechanism.** Possible reaction pathway of the carbonylative coupling reaction.

the reaction mixture. To this solution was subsequently added allylic acetate (1.0 equiv), $^t$BuNC (1.5 equiv), and Negishi reagent (1.5 equiv). The tube was equipped with a balloon filled with N$_2$ at 25 °C until complete consumption of the starting material. The mixture was added 1 M HCl aq. and stirred at room temperature for 0.25 h. The mixture was then extracted with EtOAc and separated organic layer was washed with brine, dried over anhydrous Na$_2$SO$_4$, and concentrated under reduced pressure to yield the crude product, which was purified by silica gel flash column chromatography.

## Data availability

The authors declare that all the data supporting the findings of this work are available within the article and its Supplementary Information files, or from the corresponding author upon request.

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

## Acknowledgements

This work was supported by the Youth 1000-Talent Plan Program, East China University of Science and Technology for startup funding, NSFC/China (21421004, 21702060), Shanghai Municipal Science and Technology Major Project (grant no. 2018SHZDZX03) and the Program of Introducing Talents of Discipline to Universities (B16017), and the Fundamental Research Funds for the Central Universities (WJ1814012). We thank Research Center of Analysis and Test of East China University of Science and Technology for the help on NMR analysis. Y.C. thanks Dr. Yang Yang (California Institute of Technology) in proofreading this manuscript.

## Author contributions

Y.C. conceived the project and wrote the paper with the feedback of the other authors. J.Q. and Y.C. directed the project. Y.W., C.Z. and Z.T. performed the experiments and analyzed the data. M.S. and W.H. contributed to the discussion and commented of the manuscript.

## Competing interests

The authors declare no competing interests.
