## [Peer Review File · Nature Communications]

Reviewers' comments:

Reviewer #1 (Remarks to the Author):

Regarding "Nickel-catalyzed allylic carbonylative coupling of alkyl zinc reagents with tert-butyl isocyanide"

Chen et al report a carbonylative nickel catalyzed coupling of allylic acetates and alkyl zinc reagents using tert-butyl isocyanide as a surrogate for carbon monoxide to produce β - γ -unsaturated ketones.

The reaction selectively produces the linear (E)-products in moderate to high yields. The reaction works at low temperatures (25–50°C), is ligand free, and is complete within an hour followed by hydrolysis. These features make the method attractive from a practical point of view. The presented work is systematic and clearly illustrated in the figures. It includes a broad scope of nucleophiles and electrophiles (Fig. 2 and Fig. 3), scale up, post modification and orthogonal cross coupling (Fig. 5). Additionally, they demonstrate the application of the carbonylation on biologically relevant electrophiles (Fig. 4).

The pi-allyl nickel intermediate seems plausible based on the conducted experiments in Fig. 6 a). Have you considered using the S or R enantiomer of e.g. (E)-4-phenylbut-3-en-2-yl acetate as a substrate to investigate what type oxidative addition is occurring? Is it consistent with a Tsuji-Trost type oxidative addition (S_N2)? The discussion/text concerning the experiments in Fig 6. b) is unclear and needs to be rewritten. The result of Fig. 6 c) appears odd when compared to the results reported by Ito (ref 62) and Dechert-Schmitt (ref 31), do you know why [1-(tert-butylimino)-butyl]zinc chloride is not formed under your reaction conditions? Is it "simply" due to the mild reaction conditions? If yes, please write so.

The section called "Discussion" should be "Summary" or "Conclusion". "[1-(tert-butylimino)-butyl]zinc" should be "[1-(tert-butylimino)-butyl]zinc chloride". Space missing in ref 16 "controland" should be "control and". In Fig. 1 a), b) and c) the R²-groups should be defined as alkyl.

In general, the manuscript contains many grammatical errors and some incoherent sentences.

If proper corrections are made and the above questions considered, we then recommend the manuscript be accepted for publication.

Reviewer #2 (Remarks to the Author):

This manuscript reports an interesting method for the construction of non-symmetrical ketones by hydrolysis of imines, which are built up by Ni-catalyzed Negishi coupling involving intermediate insertion of t-butylisocyanide. The manuscript is concise and describes not only the methodology but also some applications, and provides some insight into the reaction mechanism.

The reaction gives the same compounds that can be accessed by other carbonylative couplings, and the major novelties are the use of Ni, which is important, and the use of the isocyanide as a CO surrogate. This second aspect has some drawbacks, since these are not the most convenient reagents, and lower the atom economy of the process as well. The reaction shows a wide scope and solves some of the problems previously found for Pd-catalyzed reactions. However, authors do not mention that this kind of derivatives are easily formed in Stille carbonylative couplings that do not show the drawbacks of the Negishi analogs.

In any case, this is an interesting work. Nevertheless, the current state of the art in metal-catalyzed synthetic methodology is related to processes with less functionalized starting materials

(which evolve for example by C-H activation), and that render efficient processes with high atom economy. On the other hand, the reaction does not seem to show any special relevant mechanistic features, and the regio- and stereoselectivity are the expected ones. For these reasons, the interest is somewhat limited, and this precludes publication in a general science journal. It would be more suitable for a high impact chemistry journal such as JACS or ACIE.

Reviewer #1 (Remarks to the Author):

1. Chen et al report a carbonylative nickel catalyzed coupling of allylic acetates and alkyl zinc reagents using tert-butyl isocyanide as a surrogate for carbon monoxide to produce β - γ -unsaturated ketones.

The reaction selectively produces the linear (*E*)-products in moderate to high yields. The reaction works at low temperatures (25–50°C), is ligand free, and is complete within an hour followed by hydrolysis. These features make the method attractive from a practical point of view. The presented work is systematic and clearly illustrated in the figures. It includes a broad scope of nucleophiles and electrophiles (Fig. 2 and Fig. 3), scale up, post modification and orthogonal cross coupling (Fig. 5). Additionally, they demonstrate the application of the carbonylation on biologically relevant electrophiles (Fig. 4).

Response: We appreciate the reviewer's positive comments.

2. The π -allyl nickel intermediate seems plausible based on the conducted experiments in Fig. 6 a). Have you considered using the S or R enantiomer of e.g. (*E*)-4-phenylbut-3-en-2-yl acetate as a substrate to investigate what type oxidative addition is occurring? Is it consistent with a Tsuji-Trost type oxidative addition (SN2)?

Response: We appreciate for the review's suggestion. We selected (R)-1af as the starting material, the desired product 4af was obtained in 79% isolated yield with 75% ee, the configuration of desired product 4af was inversed. This result is consistent with Tamaru chemistry (J. Org. Chem., 1995, 60, 1365), which means that the initiate step most likely undergo SN2 type in the oxidative addition process. The previous literature (J. Chem. Soc., Chem. Commun., 1994, 1789; Tetrahedron Letters, 1998, 39, 601; J. Chem. Soc., Perkin Trans. 1, 1995, 2083.) also supported the formation of π -allyl nickel intermediate in the reversion way. The erosion of ee may arise from the tautomerization between imine and enamine intermediate in the aqueous work up procedure. We also performed the water work-up condition, the ee value goes down to 60%.

In the manuscript, this description has been added as Fig 6b. The content has been rewritten: “The stereochemistry of this carbonylative coupling reaction was also examined by using (*R*)-1af as the starting material (Fig. 6b): the desired product 4af was obtained in 75% ee with the inversed configuration, which indicated that the initiate step most likely undergo SN2 type in the oxidative addition process similar to the Tsuji-Trost type oxidative addition⁶⁴⁻⁶⁶. The erosion of ee may arise from the tautomerization between imine and enamine intermediate in the acidic work up procedure.”

3. The discussion/text concerning the experiments in Fig 6. b) is unclear and needs to be rewritten.

Response: The content has been rewritten:

The *tert*-butyl isocyanide possesses strong binding affinity to the nickel center, we prepared the tetrakis(*tert*-butyl isocyanide)nickel (II) perchlorate complex **8**. Under standard condition, using 0.38 equiv **8** in the absence of additional *t*BuNC delivered the desired product in 57% yield. In contrast, using 10 mol% **8** with addition of 1.5 equiv *t*BuNC increased the yield to 82%. These results suggest that the binding of multiple *t*BuNC as ligands to the nickel catalyst may proceed with ligand dissociation process in the catalytic cycles and slow down the coupling process (Fig. 6b).

4. The result of Fig. 6 c) appears odd when compared to the results reported by Ito (ref 62) and Dechert-Schmitt (ref 31), do you know why [1-(*tert*-butylimino)-butyl]zinc chloride is not formed under your reaction conditions? Is it “simply” due to the mild reaction conditions? If yes, please write so.

Response: We agree the review’s comments. High temperature is essential for carbonylative Negishi reaction at either Ito or Dechert-Schmitt’s work. In our condition, this stepwise addition sequence doesn’t work likely due to the formation at the room temperature. Most of our substrates are running at room temperature, which implies the different reaction mechanism with the Pd-catalyzed carbonylative Negishi coupling. In the text, it has been rewritten as following:

“When we preform the [1-(*tert*-butylimino)-butyl]zinc chloride, which is prepared in situ via the reaction of *t*BuNC and *n*BuLi followed by transmetallation with ZnCl₂ at room temepature, the desired product **4t** was not observed under our conditions, in contrast to the previous findings of Ito⁶² and Dechert-Schmitt³¹ that high temperature is essential for carbonylative coupling (Fig. 6c). This result highlights the distinctive mechanism in the Ni-catalyzed carbonylative Negishi coupling reaction.”

5. The section called “Discussion” should be “Summary” or “Conclusion”.

Response: *We appreciate this suggestion. We changed it in the revised manuscript.*

6. “[1-(*tert*-butylimino)-butyl]zinc” should be “[1-(*tert*-butylimino)-butyl]zinc chloride”.

Response: *We appreciate this suggestion. We changed it in the revised manuscript.*

7. Space missing in ref 16 “controland” should be “control and”.

Response: *We appreciate this suggestion. We changed it in the revised manuscript.*

In Fig. 1 a), b) and c) the R2-groups should be defined as alkyl.

Response: We appreciate this suggestion. We changed it in the revised manuscript.

8. In general, the manuscript contains many grammatical errors and some incoherent sentences.

Response: We appreciate this suggestion. Several grammatical errors and incoherent sentences has been changed as shown in below:

- (1) "Transition metal-catalyzed carbonylation with carbon nucleophiles is one of the most prominent method for construction of ketone functionality, which is the versatile motifs prevalent in organic compounds." was changed to "Transition metal-catalyzed carbonylation with carbon nucleophiles is one of the most prominent **methods to construct ketones, which are highly versatile motifs prevalent in a variety of organic compounds.**"
- (2) "Pd-catalyzed allylic reaction represents one of the most prominent carbon-carbon bond formation and is widely applied in organic chemistry," was changed to "Pd-catalyzed allylic reaction represents one of the most prominent **carbon-carbon bond forming reactions with wide synthetic applications** in organic chemistry,"
- (3) "enabling the expedient synthesis of the versatile β,γ -unsaturated ketones which are ubiquitous motifs in bioactive compounds and utilized as an valuable synthetic building blocks¹⁴⁻²⁰." was changed to "enabling the expedient synthesis of the versatile β,γ -unsaturated ketones which are **ubiquitous** motifs in bioactive compounds and utilized as valuable synthetic building blocks¹⁴⁻²⁰."
- (4) "and a mixture of the linear and branched coupling product were usually obtained when simple alkyl Negishi reagents were used (Fig. 1a)." was changed to "a mixture of the linear and branched coupling **products** were usually obtained when simple alkyl Negishi reagents were used (Fig. 1a)."
- (5) "At the outset of our investigation, we recognized that several issues need to be addressed to develop an effective nickel-catalyzed allylic carbonylation." was changed to "At the outset of our investigation, we recognized several issues **that needed** to be addressed in order to develop an effective nickel-catalyzed allylic carbonylation."
- (6) "especially to circumvent the long term limitations." Was changed to "especially to circumvent the **long-term** limitations."
- (7) "Neither the branched nor the secondary allylic products as observed in previous work²¹ was detected." was changed to "Neither the branched nor the secondary allylic products as observed in previous work²³ **were detected.**"
- (8) "Additionally, when allyl acetate containing a cyclopropyl group (1ar) was subjected to the standard conditions, ring-opening product of cyclopropyl group product could not be detected (Fig. 6a)." was changed to "Additionally, when allyl acetate containing a cyclopropyl group (1ar) was subjected to the

standard conditions, ring-opening product of cyclopropyl group could not be detected (Fig. 6a).”

9. If proper corrections are made and the above questions considered, we then recommend the manuscript be accepted for publication.

Response: We appreciate the reviewer's positive comments. Hopefully the changes made above would meet your standard.

Reviewer #2 (Remarks to the Author):

1. This manuscript reports an interesting method for the construction of non-symmetrical ketones by hydrolysis of imines, which are built up by Ni-catalyzed Negishi coupling involving intermediate insertion of t-butylisonitrile. The manuscript is concise and describes not only the methodology but also some applications, and provides some insight into the reaction mechanism.

The reaction gives the same compounds that can be accessed by other carbonylative couplings, and the major novelties are the use of Ni, which is important, and the use of the isonitrile as a CO surrogate.

Response: We appreciate the reviewer's positive comments.

2. This second aspect has some drawbacks, since these are not the most convenient reagents, and lower the atom economy of the process as well. The reaction shows a wide scope and solves some of the problems previously found for Pd-catalyzed reactions. However, authors do not mention that this kind of derivatives are easily formed in Stille carbonylative couplings that do not show the drawbacks of the Negishi analogs. In any case, this is an interesting work. Nevertheless, the current state of the art in metal-catalyzed synthetic methodology is related to processes with less functionalized starting materials (which evolve for example by C-H activation), and that render efficient processes with high atom economy. On the other hand, the reaction does not seem to show any special relevant mechanistic features, and the regio- and stereoselectivity are the expected ones. For these reasons, the interest is somewhat limited, and this precludes publication in a general science journal. It would be more suitable for a high impact chemistry journal such as JACS or ACIE.

Response:

Carbonylative cross coupling with organotin compounds indeed represent an important class of carbonylations. Stille group developed serials of Pd-catalyzed allylic carbonylative Stille cross-coupling reactions, however, the transfer group of organostannyl reagents mainly limited on the aryl, vinyl and allyl group (*JACS*, **1983**, *105*, 7173.; *JACS*, **1984**, *106*, 4833.). In addition to avoiding toxic tin reagents, we developed the highly regioselective Ni-catalyzed allylic carbonylative coupling with alkyl zinc reagents, which would be an implement with Stille's work.

In the revised manuscript, we added the comments of carbonylative Stille couplings as below: “The Stille group has realized the Pd-catalyzed allylic carbonylative Stille coupling, while the organotin reagents largely limited on aryl, vinyl and allyl stannanes²¹⁻²².” The related references are also cited.

Chemists in the pharmaceutical sciences need methodologies that are robust, scalable, and environmentally friendly. Although in principle a C-H functionalization method would be an important advance, the ability to use readily available Negishi reagents will empower researchers to immediately access new materials. This view is broadly shared by recent Ni-catalyzed Negishi cross couplings (*Nature*, **2018**, 56, 350.; *Science*, **2018**, 360, 75.; *Nature*, **2017**, 545, 213.; *Science*, **2016**, 352, 801.). Transition metal-catalyzed allylic cross coupling reactions represent one of the most important carbon-carbon bond formation reactions in organic chemistry. The formation of β , γ -unsaturated ketone in three component cross-coupling reactions possesses both alkene and ketone functionalities in a single molecular, which provide the many feasibilities for further manipulations.

REVIEWERS' COMMENTS:

Reviewer #1 (Remarks to the Author):

I have now gone through the changes made by the authors and I am fully satisfied with this and can now recommend this interesting work to be published in Nature Communications.